# Mechanical Acaricides Active against the Blacklegged Tick, *Ixodes scapularis*

**DOI:** 10.3390/insects13080672

**Published:** 2022-07-26

**Authors:** Elise A. Richardson, Loganathan Ponnusamy, R. Michael Roe

**Affiliations:** Department of Entomology and Plant Pathology, North Carolina State University, 3230 Ligon Street, Raleigh, NC 27695, USA; earicha5@ncsu.edu (E.A.R.); lponnus@ncsu.edu (L.P.)

**Keywords:** industrial minerals, Celite 610, Imergard^TM^ WP, diatomaceous earth, perlite

## Abstract

**Simple Summary:**

*Ixodes scapularis,* also known as the blacklegged tick or deer tick, is the vector of the bacteria that causes Lyme disease in humans, the most common vector-borne disease in the United States. Synthetic chemical pesticides are used to control ticks. Environmentally friendly, new methods are needed to manage chemical pesticide resistance. We evaluated the efficacy of the industrial mineral, Celite 610, an amorphous silica, against unfed *I. scapularis* nymphs. Celite is found in nature and has a mechanical, non-toxic mode of action. Dipping ticks into Celite for 1–2 s resulted in 90% mortality in as little as 69 min. Scanning electron microscopy suggested that one mode of action could be the physical obstruction of respiration. We developed another industrial mineral made from volcanic glass, Imergard^TM^ WP, for mosquito and filth fly control. In studies here, Imergard had similar activity as Celite against the deer tick. This research, although needing further study, suggests that industrial minerals could be a new, safe (“found in toothpaste”) and persistent (“it’s rock”) alternative to chemical pesticides to control ticks.

**Abstract:**

Cases of Lyme disease in humans are on the rise in the United States and Canada. The vector of the bacteria that causes this disease is the blacklegged tick, *Ixodes scapularis*. Current control methods for *I. scapularis* mainly involve chemical acaricides. Unfortunately, ticks are developing resistance to these chemicals, and more and more, the public prefers non-toxic alternatives to chemical pesticides. We discovered that volcanic glass, Imergard^TM^ WP, and other industrial minerals such as Celite 610 were efficacious mechanical insecticides against mosquitoes, filth flies, and agricultural pests. In this report, when 6–10- and 50–70-day old unfed *I. scapularis* nymphs were dipped for 1–2 s into Celite, the time to 50% mortality (LT_50_) was 66.8 and 81.7 min, respectively, at 30 °C and 50% relative humidity (RH). The LT_50_ was actually shorter at a higher 70% RH, 43.8 min. Scanning electron microscopy showed that the ticks were coated over most of their body surface, including partial to almost total coverage of the opening to their respiratory system. The other mechanical insecticide, Imergard, had similar efficacy against blacklegged unfed nymphs with an LT_50_ at 30 °C and 50% RH of 70.4 min. Although more research is needed, this study suggests that industrial minerals could be used as an alternative to chemical pesticides to control ticks and Lyme disease.

## 1. Introduction

As the planet experiences a changing climate, incidences of arthropod vector-borne disease are on the rise as the habitat of insects and ticks expands. Tick-borne diseases make up 95% of vector-borne diseases in the United States and about 70% of these cases are Lyme disease; this makes Lyme disease the most common vector-borne disease in the US [1,2,3]. Lyme disease is caused by the spirochete bacteria, *Borrelia burgdorferi,* which causes serious illness that affects humans, companion animals, and livestock. This disease is transmitted by the bite of the blacklegged tick, *Ixodes scapularis*, and is a growing threat to public health throughout the US and Canada. *Ixodes scapularis* is responsible for the transmission of other human pathogens including anaplasmosis, babesiosis, ehrlichiosis, and Powassan virus disease. The tick has an expanding geographic range, most likely due to climatic change, increasing the risk of human exposure to Lyme disease [4,5]. 

Controlling *I. scapularis* populations and preventing the public from being bitten by ticks is crucial to reducing Lyme disease and other tick-borne illnesses. There are numerous methods for preventing blood feeding on humans, i.e., repellents, permethrin-treated clothing, synthetic chemical pesticides to reduce wild tick populations on and off animals, reducing reservoir populations such as deer and rodents, and the vaccination of these reservoir hosts [1,6,7,8]. However, the majority of current tick control tactics rely heavily on synthetic chemical acaricides with few available and efficacious acaricide options derived from nature. Because the most common pesticides for vector control are used in agriculture, urban pest management, and tick control, the evolution of pest resistance to chemistry is increasing [9,10,11,12]. Numerous tick species have demonstrated resistance to chemical acaricides [13]. While this has not been reported yet in *I. scapularis*, it is possible this could occur in the future [13,14]. There is a need for environmentally friendly alternatives to permethrin (commonly used against ticks), which has been shown to have detrimental effects on non-target organisms [15]. Moreover, there is a large public demand for sustainable, safe, biorational alternatives to chemical pesticides. 

We recently discovered two novel mineral-based insecticides for use in vector control, Celite 610 and Imergard^TM^ WP. Celite is derived from a unique species of diatomaceous earth, and Imergard is 100% expanded perlite, an alumina–silica volcanic rock. Both Celite and Imergard work as mechanical insecticides, with Imergard being effective against mosquitoes and filth flies [16,17,18] and Celite against filth flies and sand flies (Roe, unpublished) and for the control of thrips in cotton [19]. Celite, under the trade name Deadzone, was also active against *A. americanum* [20]. 

The mode of action of mechanical insecticides is thought to be physical, through the abrasion of the cuticle and/or absorption of cuticular lipids, resulting in increased water loss and desiccation [21,22]. Mechanical insecticides have been shown to be effective on a variety of different types of insects such as weevils, grain beetles, whiteflies, bedbugs, cockroaches, filth flies, fleas, mosquitoes, and ticks [16,17,20,23,24,25,26]. Both perlite and diatomaceous earth-based products are non-toxic to vertebrates and occur naturally [16,20,27]. Perlite, the active ingredient in Imergard, was classified by the US Food and Drug Administration (FDA) as safe for use in juice filter aids and whitening agents in toothpaste [28,29]. Diatomaceous earth-based products such as Celite and Imergard can be stored indefinitely, are persistent, and because of their non-systemic mode of action, should not elicit cross-resistance to chemistry [20,21,30]. This study examines the efficacy and mode of action of Celite and Imergard on *I. scapularis*, the most important arthropod vector threatening human health in the US. Compared to the other life stages, the nymph was responsible for most Lyme disease cases [31]. 

## 2. Materials and Methods

### 2.1. Ticks and Industrial Minerals 

Unfed *I. scapularis* (Acari: Ixodidae) nymphs were obtained from the Oklahoma State University tick laboratory (Stillwater, OK, USA). Once received, ticks were kept at least overnight in an insect rearing room at 27 °C, 70% relative humidity, and a 14:10 h light: dark cycle to acclimate and were used within 7 d. The ticks, at the time they were received, were of two different ages: 6–10 and 50–70 d post-ecdysis to the nymphal stage. The mechanical insecticides used in these assays were Celite 610 and Imergard™ WP, which were provided by the company Imerys (Imerys Filtaration Minerals, Inc., Roswell, GA, USA.). The minerals were stored at room temperature in the dark in their original packaging containers until used. 

### 2.2. Determining the Time to Mortality of I. scapularis after Exposure to Celite 610

Nymphs were exposed to Celite at either 6–10 d at 50% relative humidity (RH), 50–70 d at 50% RH, or 50–70 d at a high humidity of 70% RH. All assays were run at 30 ± 1 °C. Each tick was dipped for 1–2 s using a round #4 camelhair paintbrush (Craft Smart, Irving, TX, USA) into a plastic Petri dish bottom containing about 25 mg of Celite in an area of about 10 square mm. Ticks were submerged, exposing all surfaces to the Celite. After each dipping, ticks were moved to a clean plastic Petri dish bottom (5 ticks per plate (Fisherbrand 60 mm × 15 mm; Fisher Scientific, Hampton, NH, USA)), and the top of the plate was used to cover the bottom plate. The two plate halves were sealed together with Parafilm (Bemis Company, Inc., Neenah, WI, USA) to prevent ticks from escaping, and the Petri dish was incubated at 30 ± 1 °C, the respective assigned RH, and a 16L:8D photoperiod. Temperature and RH were measured with a Fluke 971 (Fluke Corporation, Everett, WA, USA) temperature and humidity meter. Ticks were monitored for death through the clear Petri dish top every 10 min until 100% mortality was obtained. Ticks were considered dead if they did not move after being disturbed by gently tapping the Petri dish. Controls followed the same protocol and were handled in the same manner, except that they were dipped into an empty Petri dish. Following the assay, the Petri dishes containing ticks were stored at −40 °C. After at least 48 h, ticks were individually placed in Eppendorf tubes (Fisherbrand 1.5 mL, Fisher Scientific, Hampton, NH, USA) and stored at −40 °C. The assays with 6–10 d old ticks at 50% RH were replicated three times with each replicate containing 10–15 ticks for a total of 40 ticks used in the treatment and 40 ticks used in the control. The assay with 50–70 d old ticks at 50% RH was replicated three times with each replicate containing 15–16 ticks, for a total of 46 ticks tested and 45 ticks used in the control. The high humidity (70% RH) assays used 50–70 d old ticks and were replicated three times, with each replicate containing 10–21 ticks for a total of 46 ticks for the treated assays and 45 for the controls. Each replicate consisted of 2–4 treatment and control Petri dishes, which were completed on different days. Ticks were examined at 24 h to confirm death.

### 2.3. Determining the Time to Mortality of I. scapularis after Exposure to Imergard™ WP

Unfed nymphs were 50–70 d old, and exposure followed the same methodology as the Celite assays, except we used Imergard. After dipping, ticks were incubated at 30 ± 1 °C, 50 ± 5% RH, and a 16L:8D photoperiod. Mortality was assessed every 10 min, as described earlier. Upon reaching 100% mortality, ticks were kept in the incubator and examined at 24 h to confirm death. This assay used a total of 40 treated ticks and 40 control ticks with 5 ticks in each Petri dish.

### 2.4. Statistical Analysis

Time course data were analyzed using a Probit model using the IBM^®^ SPSS^®^ Statistics 2021 software package, version 28 (Armonk, NY, USA). The Probit model was used to calculate the lethal time to achieve 50% mortality (LT_50_), the lethal time to achieve 90% mortality (LT_90_), the 95% confidence intervals (CI), and the χ2 of the time course data. There was no control mortality, and therefore Abbot’s correction was not required. The data were normally distributed and did not require any transformation.

### 2.5. Scanning Electron Microscopy

Ticks that died by dipping into Celite and were stored without contact with each other at −40 °C were selected at random and examined by scanning electron microscopy (SEM). The selected ticks’ SEM imaging was performed at the Analytical Instrumentation Facility at North Carolina State University (NCSU). Ticks were vacuum desiccated for 48 h and then mounted with super glue onto an aluminum Hitachi SEM mount. This was followed by Cressington sputter coating for 60 s with a 70 nm gold–palladium mixture (60 Au/40 Pd). The ticks were scanned with a Hitachi SU3900 (Hitachi, Ltd., Chiyoda City, Tokyo, Japan) variable pressure scanning electron microscope to image the Celite on the tick surface.

## 3. Results

### 3.1. Efficacy of Celite 610 against I. scapularis

The structure of Celite is shown in Figure 1. Mortality was first observed after 20 min for 6–12 d old ticks and after 30 min for 50–70 d old ticks at 50% RH with 100% mortality obtained at 140 and 170 min, respectively (Figure 2). At 70% RH, the first mortality for 50–70 d old ticks was at 20 min and 100% mortality at 100 min.

Probit models were used to determine the time to 50% mortality (LT_50_) and time to 90% mortality (LT_90_) (Table 1). The LT_50_ for 6–10 d old unfed nymphs at 50% RH was 66.84 min (95% CI 62.12 to 71.43 min). The LT_90_ was 113.42 min (95% CI 106.49 to 122.12 min). Ticks 50–70 d old at the same RH had an LT_50_ of 81.68 min (95% CI 77.44 to 85.86 min) and an LT_90_ of 128.31 min (95% CI 122.26 to 135.53 min). Higher age increased the time to death at the LT_50_ and LT_90_ by 1.2- and 1.1-fold, respectively.

Ticks incubated at a higher RH died faster (Table 1). The 50–70 d old nymphs at 70% RH had an LT_50_ of 43.84 min (95% CI 40.61 to 46.93 min) and an LT_90_ of 69.46 min (95% CI 65.18 to 74.89 min) (Table 1). Increased RH decreased the time to death at both the LT_50_ and LT_90_ by 1.9- and 1.8-fold, respectively. To confirm mortality, all ticks were found to be dead at 24 h as well. The control ticks remained active and were walking around the assay arena throughout the assays, while the ticks exposed to Celite were obviously less active after dipping. They were quiescent, with some ability to move their legs while laying on their backs, and some were able to crawl.

### 3.2. Efficacy of Imergard™ WP against I. scapularis

Imergard was also active against 50–70 d old unfed nymphal *I. scapularis* ticks by dipping and incubation at 30 ± 1 °C and 50 ± 5% RH. Mortality first occurred at 20 min with 100% mortality at 160 min (Figure 3). The LT_50_ was 70.40 min (95% CI 65.99 to 74.67) and the LT_90_ was 112.56 min (95% CI 106.61 to 119.78) (Table 1). Imergard was more active than Celite on unfed, 50–70 d old nymphs at 50% RH by 1.2-fold at the LT_50_ and 1.1-fold at the LT_90_.

### 3.3. Scanning Electron Microscopy

Scanning electron microscopy was performed on ticks after exposure to Celite and their death (Figure 4). The ticks were fully covered in Celite with variations in the amount between ticks on their dorsum (Figure 4A,B) and from one place to the other on the dorsum (Figure 4B). The capitulum had a similar coating level to the other areas of the dorsum (Figure 4A,C). As shown on the dorsum in Figure 4C, high coating levels were not found on the capitulum. The venter consistently had high levels of coverage, as shown in Figure 4D, and the legs appeared not to display these high levels (Figure 4A,B,D). Figure 5A shows the spiracular plate in the control, which leads to the tracheal (respiratory) system. Figure 5B,C shows that the opening is partially or completely covered, respectively, by Celite. Additionally, no cuticle damage was observed on the dorsum of the ticks (Figure 4A).

## 4. Discussion

Celite 610 and Imergard WP demonstrated acaricidal activity against *I. scapularis* unfed nymphs in our dip submersion assay. Ticks 6–10 d past molt exposed to Celite had a shorter LT_50_ of 66.84 min compared to 50–70 d old ticks at 81.68 min (Table 1). This also occurred at the LT90. The age difference in time to death might reflect a lower water permeability across the cuticle and/or the desiccation resistance of the tick in general in the older nymphs, as older nymphs may have a more developed cuticle that provides better protection against desiccation [17,32]. On the other hand, these results were counter-intuitive since the older ticks had a longer time since their last blood meal and might be expected to be more dehydrated than the younger ticks. It is also possible between 6–10 d and 50–70 d that the latter had more time for water to move from the digestive system into the hemolymph and/or the processing of the blood was completed and the ticks were metabolically less active with reduced levels of respiration and associated water loss.

The ticks that were exposed to Celite in high humidity conditions (70% RH) had an LT_50_ of 43.83 min, which is about half the time as the ticks of the same age at 50% RH where the LT_50_ was 81.68 min (Table 1). This result was the opposite of what would be expected for a number of reasons. Normally, high humidity conditions increase tick survivability [33,34]. Ticks are at a constantly high risk of desiccation, especially when host-seeking [35], and the tick’s microclimate and especially temperature and humidity affect their locomotory activity, habitat position, and survival [33,34,36,37,38]. Berger [39] found that low relative humidity reduced *I. scapularis* activity. Ticks have also been shown to change their behavior to avoid desiccation [40,41]. In our assays after treatment, the ticks curled their legs inward and became quiescent, moving less. This could have been a behavioral attempt to prevent further dehydration. Previous studies with insects found that the efficacy of diatomaceous earth (DE) was reduced under high humidity conditions [42,43,44]. Even slight changes in relative humidity significantly affected the reduced efficacy of DE [42]. Finding the opposite effect of high humidity on ticks is not clear and, along with the rapid reduced activity and mortality after treatment with Celite and Imergard, suggests that another or additional mechanism of action other than dehydration is occurring. More research is needed to clearly understand the mode of action of these mechanical acaricides on ticks.

The mode of action of mechanical insecticides and acaricides is thought to occur by physical damage to the cuticle, leading to desiccation [21,22]. This is thought to also be the mode of action for Celite and Imergard. However, in our study, we did not observe any damage to the cuticle that could have led to desiccation in our SEM images (Figure 4 and Figure 5). Furthermore, the ticks were not physically active after treatment, suggesting that abrasion was not a factor in cuticle damage. The thick cuticle of a tick was reported to be highly water impermeable [45,46]. Additionally, an approximately two-fold reduction in the time to mortality at 70 versus 50% RH suggests that dehydration is not a mechanism of action. If the disruption of water balance was a factor, we would expect to see ticks die at a slower rate in high humidity. This led us to consider other possible modes of action. Figure 5B shows that the opening into the tracheal system was occluded by Celite, whereas in the case of Figure 5C, there appears to be an almost total blockage. The spiracles of a tick are a critical part of its respiratory system, responsible for the regulation of gas exchange but also limiting water loss from the tracheal system [47,48,49]. Other studies have found that when the spiracular plate is wet with alcohol, it leads to decreased survival in *Ixodes* ticks [46]. With this in mind, one contributing factor to mortality might be the impact of the mineral on tick respiration.

Ticks exposed to Imergard produced an LT_50_ of 70.40 min at 50% RH (Table 1). This LT_50_ was about 11 min shorter than that for Celite under the same conditions (81.68 min). This technically suggests that Imergard is more effective at killing *I. scapularis*, although practically does not appear to be of major consequence. What is more interesting is that two different mineral types, volcanic glass versus amorphous silica, respectively, are almost equally efficacious. This further supports the idea that the mode of action is physical and not systemic. Understanding the exact mode of action will be critical to studies to improve the efficacy of these minerals. One advantage of Imergard is that it does not contain crystalline silica [17], although the levels in Celite are also considered safe.

The mode of action of Imergard in insect studies so far appears to be dehydration, but how this is accomplished is still unclear. In studies with mosquitoes treated with Imergard, no cuticle abrasion was found and the majority of the mineral was found on the legs [18]. Once mosquitoes came into contact with the mineral, they were no longer active [18], similar to what was seen in this study. Chen et al. [17] concluded that filth flies exposed to Imergard died due to a disruption of water balance. In both mosquitoes and filth flies, the time to 50% mortality increased when the humidity was increased, suggesting that the mechanism of action might be different between insects and ticks.

This study is the first to test the effectiveness of the minerals Celite and Imergard on *I. scapularis*. Our results suggest that both of these minerals could be an alternative method of the control of the blacklegged tick, Lyme disease, and other tick-borne diseases. Both of the active ingredients in these products are considered generally safe [16,20,27]. Future research is needed to assess their use under more practical conditions.

## 5. Conclusions

In summary, the mineral-based products, Celite 610 and Imergard WP, were found to be efficacious against unfed nymphs of *I. scapularis* in laboratory conditions, making them promising potential acaricides. The mode of action of these industrial minerals is not clear. No damage to the cuticle was observed. However, the minerals obstructed the openings to the respiratory system. It appears these industrial minerals could be suitable alternatives to the currently used synthetic acaricides and, because of their putative mechanical mode of action, used as an additional tool for vector control and to manage tick pesticide resistance.

## Figures and Tables

**Figure 1 insects-13-00672-f001:**
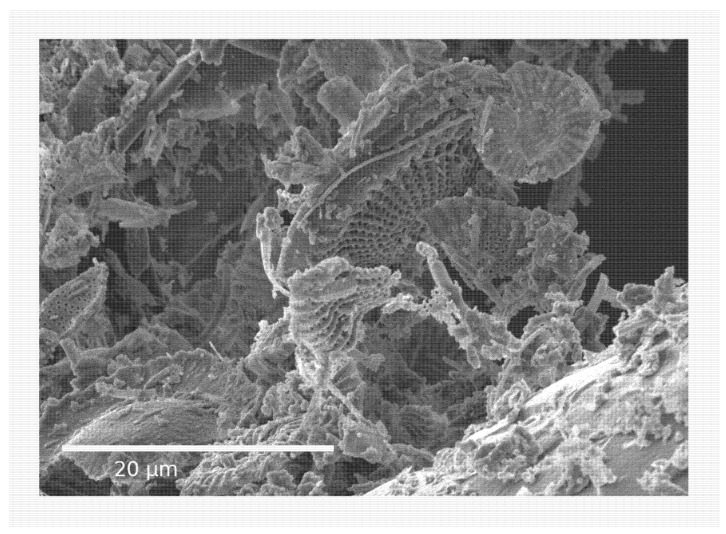
Scanning electron microscopy of Celite 610.

**Figure 2 insects-13-00672-f002:**
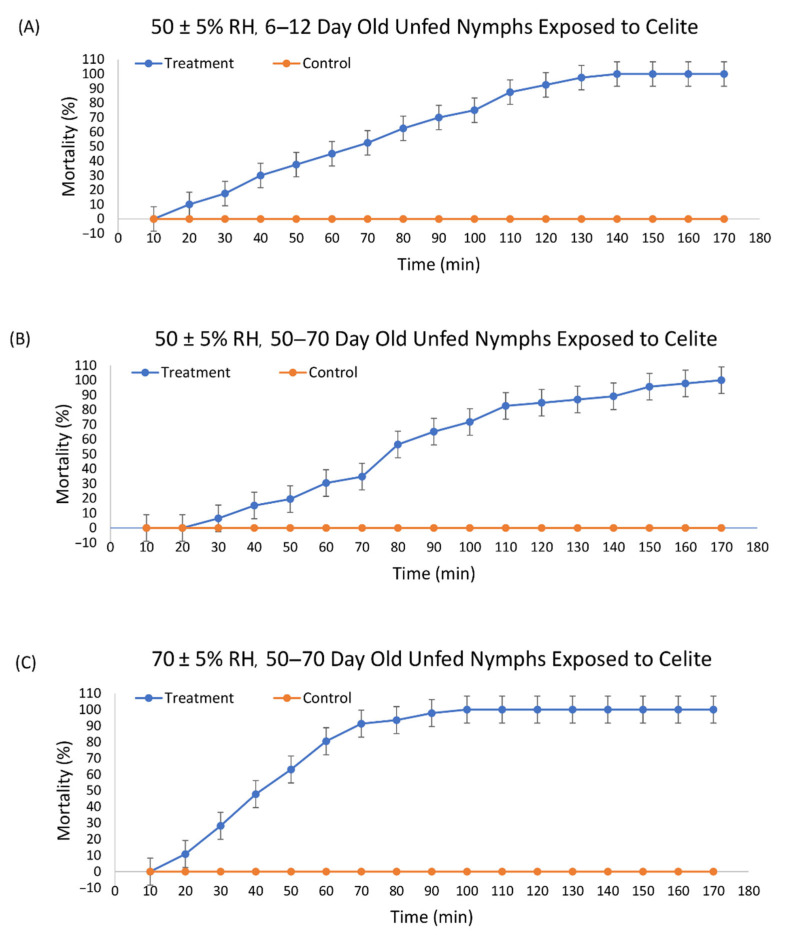
Mortality over time of unfed nymphal *Ixodes scapularis* exposed by dipping into Celite 610 with (**A**) ticks 6–10 days past last molt at 50 ± 5% relative humidity (RH), (**B**) ticks at 50–70 days past last molt at 50 ± 5% RH, and (**C**) ticks 50–70 days past last molt at 70 ± 5% RH.

**Figure 3 insects-13-00672-f003:**
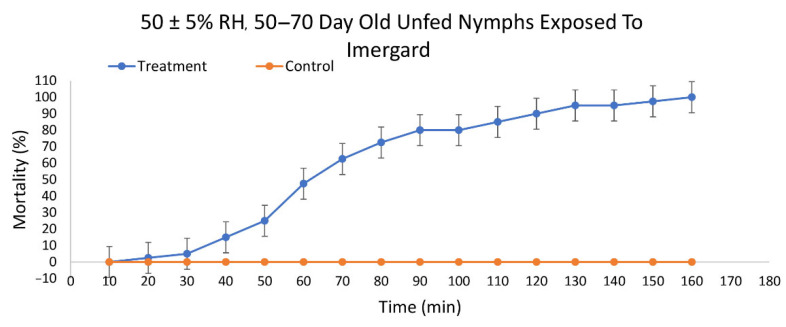
Mortality over time of unfed nymphal *Ixodes scapularis* ticks 50–70 days past last molt exposed to Imergard WP by dipping and incubated at 30 ± 1 °C and 50 ± 5% relative humidity.

**Figure 4 insects-13-00672-f004:**
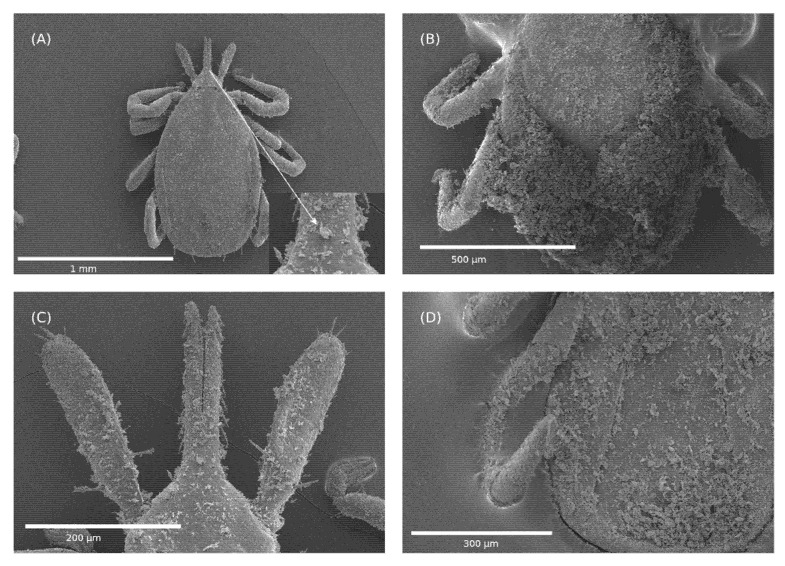
Scanning electron micrographs of *Ixodes scapularis* unfed nymphs exposed to Celite 610 by dipping. (**A**) Dorsal side of the tick, (**B**) close up of the tick on the dorsal side, (**C**) capitulum of the tick, and (**D**) ventral side of the tick, also showing legs. The arrow in (**A**) shows a close-up of a particle of Celite and reveals that there is no cuticle damage around the particle after being exposed to Celite.

**Figure 5 insects-13-00672-f005:**
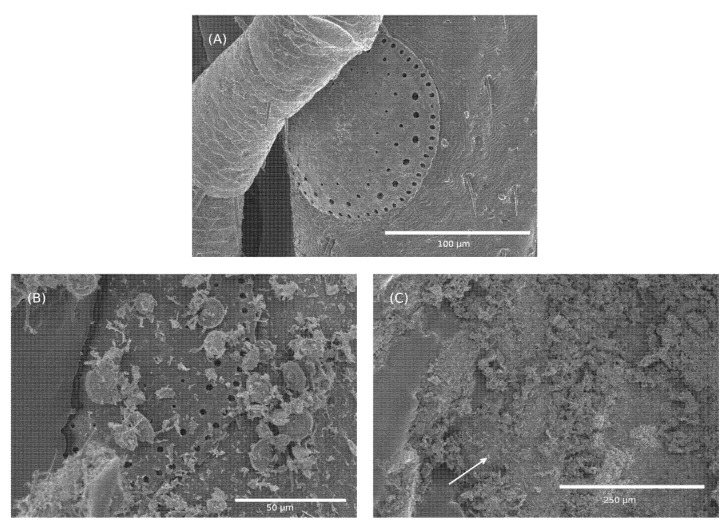
Tick spiracular plate (**A**) untreated and (**B**,**C**) treated with Celite 610. The arrow in (**C**) points to the location of the spiracular plate almost completely obstructed by Celite.

**Table 1 insects-13-00672-t001:** Probit model, LT_50_ and LT_90_ of *Ixodes scapularis* unfed nymphs exposed by dipping into Celite 610 and Imergard.

Mineral	RH *	Last Molt (Days)	N	Slope (SE)	LT_50_ ^†^ (95% CI ^‡^)	LT_90_ ^†^ (95% CI ^‡^)	χ2
Celite 610	50 ± 5%	6–10	40	0.028	66.84A (62.12, 71.43)	113.42A (106.49 122.12)	7.20
Celite 610	50 ± 5%	50–70	46	0.027	81.68B (77.44, 85.86)	128.31B (122.26, 135.53)	10.90
Celite 610	70 ± 5%	50–70	46	0.050	43.84C (40.61, 46.93)	69.46C (65.18, 74.89)	5.12
Imergard™ WP	50 ± 5%	50–70	40	0.030	70.40A (65.99, 74.67)	112.56A (106.61 119.78)	17.464

* Values indicate relative humidity, RH (%); ^†^ Values are in min; ^‡^ In each column, lethal rates with overlapping confidence intervals (CI) are not significantly different and are indicated by the same letter.

## Data Availability

Data available by request with justification from the corresponding author.

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
