# Peer review of "Mechanical Acaricides Active against the Blacklegged Tick, Ixodes scapularis"

_insects, 2022, doi:10.3390/insects13080672_

Round 1

Reviewer 1 Report

Mechanical Insecticides to Control the Blacklegged Tick, Ixodes scapularis-The Fight Against Lyme Disease.

 The authors evaluated the efficacy of Celite 610 (industrial mineral) and Imergard (amorphous silica) against unfed Ixodes scapularis nymphs. The exposure of the nymphs to the pure products produced 100% mortality in the nymphs and the LT50 and LT90 of both products were calculated. The manuscript is generally methodologically clear and well written. Although there is no doubt about the efficacy of both pure products on the mortality of I. scapularis nymphs, the use of an unconventional method for the evaluation of the products and the absence of evaluation data of different effective concentrations raise serious questions and the feasibility of the application of both products in the field for the real control of I. scapularis. I believe that complementary tests should be carried out to establish the lethal concentrations that can give certainty of the efficacy of both products.

Main criticisms:

1.- lines 98-109.- The authors used an unconventional method of exposing the nymphs to the evaluated products, they apparently immersed each nymph in the "pure" products for 1 to 2 seconds and then placed them in clean containers . Two questions arise from this essay:

a).- Why the authors did not use any of the conventional methods of evaluation of acaricides described by the WHO or by the FAO. In this regard, there are several tests that can be performed using nymphal stages, such as Open filter paper method, Limited exposure time method, Folded filter paper method or in this case perhaps the most appropriate Topical application method (WHO: Pesticides evaluation scheme. Report of the WHO Informal Consultation on the Evaluation and Testing of Insecticides. 1996; Al-Rajhy et al. Pest Manag Sci 59:1250–1254, 2003; Elmhalli et al. Exp Appl Acarol 48, 251–262, 2009). These methods are designed mainly to determine effective concentrations by simulating the conditions of exposure of ticks to acaricides in the field, and also allow comparison of the efficacy of determined acaricides in different places or even between species of arthropods.

b).- Although the time-dependent response in a biological variable demonstrates the specificity of the effect produced (in this case by both products), this does not resolve more interesting questions for practical application. In general, any substance in nature can have a potential toxic effect on an organism, what is required is exposure to a certain concentration. Under this angle, the death of nymphs due to their exposure to a highly concentrated product (close to 100%), may be a matter of time. Similar effects could be observed with soap or quicklime.

2.- The most interesting question about any potential acaricidal substance is its effective concentration, this allows us to know if the product in question really has a good chance of being used without causing toxic effects in mammals or significant damage to the environment (ecotoxicity). Again, conventional tests can resolve this issue. In this case, assuming that any of the evaluated products could be used as an acaricide in the field, what could be its use strategy, in pasture or in susceptible animals? Sprinkling or immersion? At what concentration? Highly concentrated products that have been used as acaricides are organic oils, creosote, kerosene, and tobacco, their concentration in the field did not exceed 10%. Currently conventional acaricides act at a rate of mg/ml or in the order of parts per million (ppm), none known is used in pure form. So, assuming that the products evaluated in this work can be used, what would be the cost if they must be used at a very high concentration?

Lines 2-3.- Title: "Mechanical Insecticides to Control the Blacklegged Tick, Ixodes scapularis-The Fight Against Lyme Disease"

The term "mechanical insecticide" is not common, this reviewer did not find bibliographic support about a group of substances that fit this concept. A pesticides are any substance or mixture of substances of chemical or biological ingredients intended for repelling, destroying or controlling any pest, or for regulating plant growth (FAO). In this case, the observed effect is on ticks, so a more appropriate term may be acaricide.

On the other hand, it is true that Ixodes scapularis is an important vector for Lyme Disease and other diseases, however, the results of the study are very preliminary and their use as possible acaricides is speculative in the first place, no trial was carried out on the effect of the products evaluated on the transmission capacity of some pathogen, so their use and possible impact for the control of Lyme Disease is speculative in a second order (speculation of speculation). Therefore, Lyme Disease should be removed from the title and its argument in the introduction and conclusions should also be reduced or eliminated.

Minor criticisms:

1.- Keywords.- Eliminate Lyme Disease.

2.- Figures 1, 3a and 4b.- Eliminate decimals from the scale bar.

3.- Lines 212-213.- “Celite 610 and Imergard WP demonstrated acaricidal activity against I. scapularis un-fed nymphs in our dip submersion assay, resulting in 100% mortality in <170 min….”

This occurred in the Celite 610 exposure at 50% RH but at 70% RH 100% mortality was observed at 100 mins, in the case of Imergard 100% mortality was observed at 160 mins, I suggest rephrasing .

4.- Insecticide, acaricide or pesticide? I suggest standardizing the term throughout the text.

Author Response

Reviewer 1:

  1. The authors evaluated the efficacy of Celite 610 (industrial mineral) and Imergard (amorphous silica) against unfed Ixodes scapularis nymphs. The exposure of the nymphs to the pure products produced 100% mortality in the nymphs and the LT50 and LT90 of both products were calculated. The manuscript is generally methodologically clear and well written. Although there is no doubt about the efficacy of both pure products on the mortality of I. scapularis nymphs, the use of an unconventional method for the evaluation of the products and the absence of evaluation data of different effective concentrations raise serious questions and the feasibility of the application of both products in the field for the real control of I. scapularis. I believe that complementary tests should be carried out to establish the lethal concentrations that can give certainty of the efficacy of both products.

Authors’ response: Thank you for taking the time to review our manuscript and provide useful feedback. Semi-field and field studies as you indicated are needed to assess how our minerals will be used in tick control in the future; and ultimately how much transfers to the tick to produce a mortal response is where the “rubber meets the road”.  However, dose response from topical applications is not straightforward or easy to conduct as you might think, with a non-soluble, mineral with a non-systemic mode of action and where the site of action on the tick surface is unknown. It is not just a matter of how much is applied per tick, but where it is applied and what carrier do you use and what method can be developed to apply the mineral to specific areas of the body. This is a whole project in itself, and we agree important in understanding our technology.

           Dose response is more complicated than what was just mentioned even, and we want to expand our response on this issue. We have conducted dose response studies in semi-field experiments using a modified WHO cone test on multiple species of filth flies (research cited in our tick paper under review). Dose is important in the practical use of Celite and Imergard mostly in terms of the duration of control from a treated surface.  Essentially, our data showed with filth flies that an increased dose below a low threshold did not increase the mortality rate. Treatment level on substrates were important, however, to achieve exceptional long-term control. For example, we achieve 11 months of control for mosquitoes in 4000 homes in Africa at an application rate of 5 g per square meter of wall space. Treatment level is also commercially and practically not an issue because of the low cost of the product, its non-toxic mode of action on animals and it sticks onto surfaces until an insect touches it. 

           This paper is a proof-of-concept study to determine if the minerals work against ticks, but not just any tick, but the most important disease-causing tick in North America. We suggest our data were successful in establish this proof, which justifies future work on how to use the technology on ticks in the field. We just received a grant to expand our research based on the data in this paper. We do not have all of the answers yet. The reviewer is correct, dose-response studies are needed at several levels as just discussed to fully understanding mode of action.  There is additional information on this subject that follows.

 Main criticisms:

  1. lines 98-109.- The authors used an unconventional method of exposing the nymphs to the evaluated products, they apparently immersed each nymph in the "pure" products for 1 to 2 seconds and then placed them in clean containers. Two questions arise from this essay:

Why the authors did not use any of the conventional methods of evaluation of acaricides described by the WHO or by the FAO. In this regard, there are several tests that can be performed using nymphal stages, such as Open filter paper method, Limited exposure time method, Folded filter paper method or in this case perhaps the most appropriate Topical application method (WHO: Pesticides evaluation scheme. Report of the WHO Informal Consultation on the Evaluation and Testing of Insecticides. 1996; Al-Rajhy et al. Pest Manag Sci 59:1250–1254, 2003; Elmhalli et al. Exp Appl Acarol 48, 251–262, 2009). These methods are designed mainly to determine effective concentrations by simulating the conditions of exposure of ticks to acaricides in the field, and also allow comparison of the efficacy of determined acaricides in different places or even between species of arthropods.

Authors’ response: While the authors acknowledge that these other suggested testing techniques are available and legitimate, we suggest that the immersion/dipping technique is commonly used and published abundantly. We provided a list below of 8 publications that used or recommended the use of dipping for testing insecticide efficacy.  A total list of papers using dipping methods would be much larger. We also included in the list a link that refers to dipping/immersion tests as a common method to assess insecticide activity (Yan Zhu 2008). The paper the reviewer actually cited justifying we use a different test, i.e., Al-Rajhy et al., Pest Manag Sci 59:1250–1254, 2003, also includes dipping as a method to assess pesticide efficacy.

           Our laboratory recently applied and was granted a U.S. patent for head lice control (US 2019 / 0166830 A1) where the dipping method was used to establish efficacy. The dipping method was the only method conducted to demonstrate proof of concept for a new commercial product for head lice egg control. The patent was examined by the US patent office, and commercial exclusivity was granted solely based on dipping data. This patent was for the use of our minerals for head lice control. 

           Other tests were suggested by the reviewer and again are reasonable tests to evaluate pesticides, but we decided not to use these tests in a proof-of-concept effort. Why? The transfer from a treated surface at least in mosquitoes where we have conducted most of our research, shows that the transfer of the minerals is most likely due to electrostatics. The type of surface affects the electrostatics. We know the transfer occurs on cement, wood, glass, and mud from our research on mosquitoes, but we have no idea if it works on filter paper, suggested as a test method by the reviewer. Then what filter paper type should we test? How is filter paper type relevant to the practical use of our minerals for tick control? The most direct approach to understand if the minerals can kill a deer tick, is to dip the tick and see what happens. 

           Another suggestion made by the reviewer is topical applications. This is a reasonable suggestion for a chemical pesticide that dissolves in water or an organic solvent, can translocate, and has a systemic mode of action. But our minerals do not dissolve in solvents, and they do not translocate. Furthermore, topical water applications in suspension can inactivate the minerals depending on the method of application.  Typical methods to topically apply chemical insecticides to an insect or tick in water does not work with our minerals because the water and minerals just roll off the animal.  Minerals in suspension in water when dry from a pool also form a strong solid structure and are inactive. The minerals can be applied in water as a wettable powder using conventional pump sprayers, however, and we showed this worked on 4000 homes in Africa achieving 11 months of mosquito control. Developing standard spray methods in the lab are more challenging than just dipping, and for this reason we used dipping.

           This paper on ticks is a proof of concept—can our minerals kill the deer tick.  The dipping method shows the minerals are active against the deer tick, and the dipping method is a well-recognized method to evaluate pesticide activity in the scientific literature. The information obtained in this study shows that these minerals have acaricidal activity.

(1) Yan Zhu 2008- https://link.springer.com/referenceworkentry/10.1007/978-1-4020-6359-6_1541

(2) Showler, A.T.; Flores, N.; Caesar, R.M.; Mitchel, R.D.; De León, A.A.P. Lethal effects of a commercial diatomaceous earth dust product on Amblyomma americanum (Ixodida: Ixodidae) larvae and nymphs. J. Med. Entomol. 2020, 57, 1575–1581, doi:10.1093/jme/tjaa082.

(3) Allan T Showler, Weste L A Osbrink, Eduardo Munoz, Ryan M Caesar, Veronica Abrigo, Lethal Effects of Silica Gel-Based CimeXa and Kaolin-Based Surround Dusts Against Ixodid (Acari: Ixodidae) Eggs, Larvae, and Nymphs, Journal of Medical Entomology, Volume 56, Issue 1, January 2019, Pages 215–221, https://doi.org/10.1093/jme/tjy152

(4) Busvine, J.R. (1946), The comparative toxicity of various contact insecticides to the louse (Pediculus humanus L.) and the bed-bug (Cimex lectularius L.). Annals of Applied Biology, 33: 271-279. https://doi.org/10.1111/j.1744-7348.1946.tb06313.x

(5) Allan T Showler, Jessica L Harlien, Lethal Effects of Commercial Kaolin Dust and Silica Aerogel Dust With and Without Botanical Compounds on Horn Fly Eggs, Larvae, Pupae, and Adults in the Laboratory, Journal of Medical Entomology, Volume 59, Issue 1, January 2022, Pages 283–290, https://doi.org/10.1093/jme/tjab140

(6) David, W., & Gardiner, B. (1950). Factors influencing the Action of Dust Insecticides. Bulletin of Entomological Research, 41(1), 1-61. doi:10.1017/S0007485300027474

(7) Allan T Showler, Bailee N Dorsey, Ryan M Caesar, Lethal Effects of a Silica Gel + Thyme Oil (EcoVia) Dust and Aqueous Suspensions on Amblyomma americanum (Ixodida: Ixodidae) Larvae and Nymphs, Journal of Medical Entomology, Volume 57, Issue 5, 1 September 2020, Pages 1516–1524, https://doi.org/10.1093/jme/tjaa054

(8) Paramasivam, M., & Selvi, C.T. (2017). Laboratory bioassay methods to assess the insecticide toxicity against insect pests-A review. Journal of entomology and zoology studies, 5, 1441-1445.

(9) Showler, A.T.; Saelao, P. Integrative Alternative Tactics for Ixodid Control. Insects 2022, 13, 302. https://doi.org/10.3390/insects13030302

  1. Although the time-dependent response in a biological variable demonstrates the specificity of the effect produced (in this case by both products), this does not resolve more interesting questions for practical application. In general, any substance in nature can have a potential toxic effect on an organism, what is required is exposure to a certain concentration. Under this angle, the death of nymphs due to their exposure to a highly concentrated product (close to 100%), may be a matter of time. Similar effects could be observed with soap or quicklime.

Authors’ response: While we agree that dose response is a common method to assess the level of toxicity for chemical insecticides and that there are reasons to look at dose even for our minerals, the proof of principle for a mechanical insecticide and the developing and having regulatory approval even for a new product, does not require dose response. This issue of the importance of dose was discussed in part earlier in our responses to this reviewer.

           However, time to death is also a measure of relative toxicity, i.e., after exposure the shorter the time to death, the greater the toxicity. This principle is well established in the insecticide scientific literature and commonly used especially for the rapid assessment of insecticide susceptibility as one example. If compound A kills in 2 sec and compound B in 10 days, it would not be uncommon for someone to conclude that compound A was more toxic. This does not preclude that both compounds might have a practical use and even maybe for the same application. 

           Our lab has published three studies with Imergard and its efficacy against mosquitoes and multiple species of filth flies using time-dependent responses, without dose response studies. In both of these papers, the reviewers had no concerns about using time to death as a measure of the efficacy of killing. We also have some minerals from diatoms that never will kill mosquitoes, even after days. 

           We have completed WHO Phase I, II and III studies on Imergard which is now in review by the WHO for use in Africa for mosquito control as a residual wall spray. In these studies, there is not a single dose response study. The lack of dose-response so far was not an issue requiring follow up information or a requirement for approval. We also just received funding from the Department of Defense this year for further advancement of these minerals as pesticides and other uses to prevent mosquito biting, and we were not asked to show any experiments assessing dose response. Although dose-response is important and we hope to provide data in this respect in future work as already presented in this response to the reviewer, we suggest here these data are not needed to establish proof of concept.

  1. The most interesting question about any potential acaricidal substance is its effective concentration, this allows us to know if the product in question really has a good chance of being used without causing toxic effects in mammals or significant damage to the environment (ecotoxicity). Again, conventional tests can resolve this issue. In this case, assuming that any of the evaluated products could be used as an acaricide in the field, what could be its use strategy, in pasture or in susceptible animals? Sprinkling or immersion? At what concentration? Highly concentrated products that have been used as acaricides are organic oils, creosote, kerosene, and tobacco, their concentration in the field did not exceed 10%. Currently conventional acaricides act at a rate of mg/ml or in the order of parts per million (ppm), none known is used in pure form. So, assuming that the products evaluated in this work can be used, what would be the cost if they must be used at a very high concentration?

Authors’ response: Celite 610 is a diatomaceous earth product, and Imergard is made of perlite, which is amorphous volcanic glass; both are naturally occurring and reported to be non-toxic to vertebrates. Dry desiccant dusts have been used in the past by being applied to vegetation, applied to animals, and by using mechanical dust applicators (Showler et al. 2018, Showler et al. 2020, Hayes et al. 1997, Loftin & Corder 2009, Swiger 2012). The cost of Celite 610 and Imergard are expected to be very affordable as they are naturally occurring (earth-based) and easy to obtain. The full references cited are below.

           Imergard testing (Phase I, II and III) have been completed for mosquito control in homes in Africa.  In Phase III studies, 4000 homes were tested where Imergard was applied as a wettable powder using standard pump sprays and 11 months of mosquito control was found. In comparison in Phase II studies, pyrethroids never achieved WHO acceptable control levels even just after application. The expected cost for Imergard is one-third the cost of chemistry. We also tested Celite in cotton for thrips control as a wettable powder using pump spraying and achieved control levels similar to Orthene, commonly used for thrips control on this crop. Celite is also used as a straight powder for beetle control in the poultry industry, and R.M.Roe has a new patent granted on the use of these minerals to control all stage of head lice with one use as a dry shampoo (the technology was licensed to a fortune 500 company).  We also have new IP for the use of these minerals for tick control and just received major funding to develop a commercial product.  So we are suggesting here based on our work and success so far, our discoveries are reasonably and commercially viable. 

(1) Showler, A. T., W. L. A. Osbrink, E. Munoz, R. M. Caesar, and V. Abrigo. 2018. Lethal effects of silica gel-based CimeXa and kaolin-based Surround dusts against ixodid (Acari: Ixodidae) eggs, larvae, and nymphs. J. Med. Entomol.

(2) Showler, A.T.; Flores, N.; Caesar, R.M.; Mitchel, R.D.; De León, A.A.P. Lethal effects of a commercial diatomaceous earth dust product on Amblyomma americanum (Ixodida: Ixodidae) larvae and nymphs. J. Med. Entomol. 2020, 57, 1575–1581, doi:10.1093/jme/tjaa082.

(3) Hayes, B. W., M. J. Janes, and D. W. Beardsley. 1972. Dust bag treatments in improved pastures to control horn flies and cattle grubs. J. Econ. Entomol. 65: 1368–1371.

(4) Loftin, K. M., and R. F. Corder. 2009. Controlling horn flies on cattle. FSA7031, University of Arkansas, Little Rock, AR. https://www.uaex.edu/ publications/PDF/FSA-7031.pdf.

(5) Swiger, S. L. 2012. Managing external parasites of Texas cattle. Texas A&M University, College Station, TX E-570. https://agrilifecdn.tamu.edu/ livestockvetento/files/2010/10/Managing-External-Parasites-of-Texas- Cattle.pdf.

  1. Lines 2-3.- Title: "Mechanical Insecticides to Control the Blacklegged Tick, Ixodes scapularis-The Fight Against Lyme Disease"

The term "mechanical insecticide" is not common, this reviewer did not find bibliographic support about a group of substances that fit this concept. A pesticides are any substance or mixture of substances of chemical or biological ingredients intended for repelling, destroying or controlling any pest, or for regulating plant growth (FAO). In this case, the observed effect is on ticks, so a more appropriate term may be acaricide.

On the other hand, it is true that Ixodes scapularis is an important vector for Lyme Disease and other diseases, however, the results of the study are very preliminary and their use as possible acaricides is speculative in the first place, no trial was carried out on the effect of the products evaluated on the transmission capacity of some pathogen, so their use and possible impact for the control of Lyme Disease is speculative in a second order (speculation of speculation). Therefore, Lyme Disease should be removed from the title and its argument in the introduction and conclusions should also be reduced or eliminated.

Authors’ response: We agree. You have made a good point, and we have removed Lyme Disease from the title and keywords. However, we have left the suggestion in the introduction and conclusion that it has the potential applications to reduce I. scapularis populations and possibly reduce cases of Lyme Disease as a result of this reduction in population; we found these minerals to be effective at killing this species of tick. We made clear that this could be a possible result of this control tactic.

           We have found the term “mechanical insecticide” to be a common usage for a group of pesticides with a specific mode of action that is physical and non-chemical.  We have cited publications below that have used the term and give some examples including a website for a company that makes mechanical insecticides. However, we have changed the title to “mechanical acaricides” as suggested instead of “insecticides,” as this is a good point, we only tested these minerals on ticks in this study.

(1) https://www.imerys-performance-minerals.com/mechanical-insecticides

(2) Mitchell, R.D.; Mott, D.W.; Dhammi, A.; Reisig, D.D.; Roe, R.M.; Stewart, D.A. Field evaluation of a new thrips control agent for coton: a mechanical insecticide. In Proceedings of the Beltwide Cotton Conference; San Antonio, 2018; pp. 786–795.

(3) Chen, K.; Deguenon, J.M.; Cave, G.; Denning, S.S.; Reiskind, M.H.; Watson, D.W.; Stewart, D.A.; Gittins, D.; Zheng, Y.; Liu, X.; et al. New Thinking for filth fly control: residual, non-chemical wall spray from volcanic glass. Med. Vet. Entomol. 2021, 35, 451–461, doi:10.1111/mve.12521.

(4) Deguenon, J.M.; Azondekon, R.; Agossa, F.R.; Padonou, G.G.; Anagonou, R.; Ahoga, J.; N’dombidje, B.; Akinro, B.; Stewart, D.A.; Wang, B.; et al. Imergard TM WP: a non-chemical alternative for an indoor residual spray, effective against pyrethroid-resistant Anopheles gambiae (s.l.) in Africa. Insects 2020, 11, 322, doi:10.3390/INSECTS11050322.

(5) Deguenon, J.M.; Riegel, C.; Cloherty-Duvernay, E.R.; Chen, K.; Stewart, D.A.; Wang, B.; Gittins, D.; Tihomirov, L.; Apperson, C.S.; McCord, M.G.; et al. New mosquitocide derived from volcanic rock. J. Med. Entomol. 2021, 58, 458–464, doi:10.1093/jme/tjaa141.

(6) Hosseini, S.A.; Bazrafkan, S.; Vatandoost, H.; Abaei, M.R.; Ahmadi, M.S.; Tavassoli, M.; Mansoreh Shayeghi, R. The insecticidal effect of diatomaceous earth against adults and nymphs of Blattella germanica. Asian Pac. J. Trop. Biomed. 2014, 4, S228, doi:10.12980/APJTB.4.2014C1282.

(7) Pesticide Information Leaflet: Mode of Action of Insecticides and Related Pest Control Chemicals for Production Agriculture, Ornamentals, and Turf. University of Maryland Extension. http://pesticide.umd.edu/uploads/1/3/5/6/13565116/pil43_modeofaction-agornamturf_2005-2013.pdf

Minor criticisms:

  1. - Eliminate Lyme Disease.

Authors’ response: Change made as suggested.

  1. Figures 1, 3a and 4b.- Eliminate decimals from the scale bar.

Authors’ response: Change made as suggested.

  1. Lines 212-213.- “Celite 610 and Imergard WP demonstrated acaricidal activity against I. scapularis un-fed nymphs in our dip submersion assay, resulting in 100% mortality in <170 min….”

This occurred in the Celite 610 exposure at 50% RH but at 70% RH 100% mortality was observed at 100 mins, in the case of Imergard 100% mortality was observed at 160 mins, I suggest rephrasing.

Authors’ response: This part of the sentence has now been deleted since these results were already discussed in the results section. The sentence now reads: “Celite 610 and Imergard WP demonstrated acaricidal activity against I. scapularis unfed nymphs in our dip submersion assay.”

  1. Insecticide, acaricide or pesticide? I suggest standardizing the term throughout the text.

Authors’ response: We have changed the title to say “acaricide” instead of “insecticide.” We have also now gone through the text and changed “insecticide” or “pesticide” to acaricide when appropriate. However, we have multiple papers also showing these minerals kill insects. 

Reviewer 2 Report

The paper presents very well in the introduction the relevance of the topic addressed, portraying the importance of the tick I. scapularis to human health, the importance of its control and the problem regarding the impact of the inadequate and exclusive use of chemical acaricides on the environment, and in human and animal health. It then suggests the use of minerals (mechanical insecticides) as a possible alternative to chemical acaricides, presenting studies with minerals in the control of arthropods and their “relative” safety.

Then, they aim to evaluate the effectiveness of Celite and Imergard on unfed nymphs of I. scapularis. I think that the investigation of the mechanism of action of minerals on ticks could be added as an objective of the study, since the SEM was performed to observe Celite on the nymphs, and this topic (mechanism of action) was widely addressed in the discussion.

Regarding the methodology, it was adequate to achieve the objective, but it needs to be clarified in some important points, such as:

The concentration of Celite and Imegard used per nymph. This point is important in terms of comparison with the concentrations used/applied for other arthropods or even other tick species, since there is a difference in susceptibility. It is also important for determining a treatment protocol and evaluating the feasibility of application.

Clarify the number of repetitions of the bioassays, fulfilling the basic concept of repeatability of the scientific method.

Regarding the results, this is the point that I believe needs more attention.

The first SEM figure needs to be further explained or removed from the paper.

The other microscopy images have misconceptions that need to be corrected (explained in detail in the attached file).

I also think it is important, in the description of the images, to address the visualization or not of lesions/corrosions in the ticks' cuticle, since this is a subject that is often addressed in the discussion.

Regarding the results of the bioassay/virulence on nymphs, the authors mainly describe the numerical values ​​to assess/show the effectiveness of Celite and Imegard on nymphs of different ages and kept under different humidity levels. I think statistical analysis should be prioritized/ in the description of the results, including in the discussion and conclusion.

The order in which the results are presented can be improved for better understanding by the reader.

The discussion was well thought out, but could be improved in some points, as suggested in the attached file.

The conclusion meets the objective of evaluating the effectiveness of Celite and Imegard on the tick I. scapularis, but I suggest adding the observations about the mechanism of action obtained in the study.

Author Response

Reviewer 2:

Main Comments/ Suggestions:

  1. The paper presents very well in the introduction the relevance of the topic addressed, portraying the importance of the tick  scapularisto human health, the importance of its control and the problem regarding the impact of the inadequate and exclusive use of chemical acaricides on the environment, and in human and animal health. It then suggests the use of minerals (mechanical insecticides) as a possible alternative to chemical acaricides, presenting studies with minerals in the control of arthropods and their “relative” safety.

Authors’ response: Thank you! We appreciate your comments and are very excited to have our research published in The Insects!

  1. Then, they aim to evaluate the effectiveness of Celite and Imergard on unfed nymphs of  scapularis. I think that the investigation of the mechanism of action of minerals on ticks could be added as an objective of the study, since the SEM was performed to observe Celite on the nymphs, and this topic (mechanism of action) was widely addressed in the discussion.

Authors’ response: This objective of the investigation of the mechanism of action of the minerals has now been added.

  1. Regarding the methodology, it was adequate to achieve the objective, but it needs to be clarified in some important points, such as:

The concentration of Celite and Imegard used per nymph. This point is important in terms of comparison with the concentrations used/applied for other arthropods or even other tick species, since there is a difference in susceptibility. It is also important for determining a treatment protocol and evaluating the feasibility of application.

Authors’ response: Ticks were dipped into 100% pure Celite or Imergard. It is difficult to quantify the amount of mineral needed on the tick surface to cause a mortal response.  However, the same technique and concentration was used for all ticks. Studies that have used these minerals on other arthropods have also used 100% pure mineral but used a different technique to apply the treatment. This is also discussed in a comment below in more detail. We also have cited below other studies that have used an immersion/dip technique to evaluate the efficacy of insecticides where this approach does not provide information on the amount of transfer.

            It is not easy to determine the amount of mineral on the surface of a tick as you might think. The mineral is non-soluble and has a non-systemic mode of action. The site of action on the tick surface is unknown. It is not just a matter of how much is applied to a tick, but where it is applied and what carrier do you use and what method can be developed to apply the mineral to specific areas of the body in order to determine the minimum amount needed to kill. Because of the low levels and density of the minerals on the tick, a gravimetric approach is not easy. We would need to label the minerals with a tracer (which changes its physical properties), develop mass spectrophotometric or flame photometric methods for mineral detection, and develop ways of removing the minerals in toto from an insect surface. This is a whole project in itself, and we agree important in fully understanding our technology. But this paper is a proof of concept. 

(1) Yan Zhu 2008- https://link.springer.com/referenceworkentry/10.1007/978-1-4020-6359-6_1541

(2) Showler, A.T.; Flores, N.; Caesar, R.M.; Mitchel, R.D.; De León, A.A.P. Lethal effects of a commercial diatomaceous earth dust product on Amblyomma americanum (Ixodida: Ixodidae) larvae and nymphs. J. Med. Entomol. 2020, 57, 1575–1581, doi:10.1093/jme/tjaa082.

(3) Allan T Showler, Weste L A Osbrink, Eduardo Munoz, Ryan M Caesar, Veronica Abrigo, Lethal Effects of Silica Gel-Based CimeXa and Kaolin-Based Surround Dusts Against Ixodid (Acari: Ixodidae) Eggs, Larvae, and Nymphs, Journal of Medical Entomology, Volume 56, Issue 1, January 2019, Pages 215–221, https://doi.org/10.1093/jme/tjy152

(4) Busvine, J.R. (1946), The comparative toxicity of various contact insecticides to the louse (Pediculus humanus L.) and the bed-bug (Cimex lectularius L.). Annals of Applied Biology, 33: 271-279. https://doi.org/10.1111/j.1744-7348.1946.tb06313.x

(5) Allan T Showler, Jessica L Harlien, Lethal Effects of Commercial Kaolin Dust and Silica Aerogel Dust With and Without Botanical Compounds on Horn Fly Eggs, Larvae, Pupae, and Adults in the Laboratory, Journal of Medical Entomology, Volume 59, Issue 1, January 2022, Pages 283–290, https://doi.org/10.1093/jme/tjab140

(6) David, W., & Gardiner, B. (1950). Factors influencing the Action of Dust Insecticides. Bulletin of Entomological Research, 41(1), 1-61. doi:10.1017/S0007485300027474

(7) Allan T Showler, Bailee N Dorsey, Ryan M Caesar, Lethal Effects of a Silica Gel + Thyme Oil (EcoVia) Dust and Aqueous Suspensions on Amblyomma americanum (Ixodida: Ixodidae) Larvae and Nymphs, Journal of Medical Entomology, Volume 57, Issue 5, 1 September 2020, Pages 1516–1524, https://doi.org/10.1093/jme/tjaa054

(8) Paramasivam, M., & Selvi, C.T. (2017). Laboratory bioassay methods to assess the insecticide toxicity against insect pests-A review. Journal of entomology and zoology studies, 5, 1441-1445.

(9) Showler, A.T.; Saelao, P. Integrative Alternative Tactics for Ixodid Control. Insects 2022, 13, 302. https://doi.org/10.3390/insects13030302

  1. Clarify the number of repetitions of the bioassays, fulfilling the basic concept of repeatability of the scientific method.

Authors’ response: This has now been clarified in the text (Lines 115-120). Every assay was replicated three times with an equal (+/- 1) number of ticks in both the controls and treatments. Each replicate contained between 2-4 Petri dishes (containing 5-6 ticks in each), and each replicate was conducted on different days.

  1. Regarding the results, this is the point that I believe needs more attention.

The first SEM figure needs to be further explained or removed from the paper.

Authors’ response: The purpose of Figure 1 was to show the composition of Celite itself, not what is looks like when dispersed on the surface of the tick.

  1. The other microscopy images have misconceptions that need to be corrected (explained in detail in the attached file).

Authors’ response: These misconceptions that were mentioned in the attached file have now been addressed below and corrected in the text.

  1. I also think it is important, in the description of the images, to address the visualization or not of lesions/corrosions in the ticks' cuticle, since this is a subject that is often addressed in the discussion.

Authors’ response: We have now added a close-up image to figure 4A to show that there is no damage to the tick cuticle around a particle of the mineral. We have also now added this information in the figure caption and in the results.

  1. Regarding the results of the bioassay/virulence on nymphs, the authors mainly describe the numerical values ​​to assess/show the effectiveness of Celite and Imegard on nymphs of different ages and kept under different humidity levels. I think statistical analysis should be prioritized/ in the description of the results, including in the discussion and conclusion.

Authors’ response: We have gone through and corrected any instances where it seemed that the statistical analysis was not prioritized. Each individual instance was addressed below in the point-by-point comments.

  1. The order in which the results are presented can be improved for better understanding by the reader.

Authors’ response: The results have now been reorganized as suggested.

  1. The discussion was well thought out, but could be improved in some points, as suggested in the attached file.

Authors’ response: All suggestions were addressed and corrected below in the comments from the attached file.

  1. The conclusion meets the objective of evaluating the effectiveness of Celite and Imegard on the tick I. scapularis, but I suggest adding the observations about the mechanism of action obtained in the study.

Authors’ response: These observations have now been added to the conclusion.

In Text Comments/ Suggestions:

  1. I suggest using keywords other than the title words (Line 34)

Authors’ response: Change made as suggested.

  1. ??? - acrines (line 39)

Authors’ response: Changed to “ticks”.

  1. What does "relatively safe" mean? (Line 76)

Authors’ response: “relatively safe” has been deleted to avoid being vague.

  1. Is the concentration in these cases similar, higher or lower than the concentration used for the control of arthropods? (Line 78-79)

Authors’ response: We believe that the concentration of perlite used in the referenced published papers regarding toothpaste whitener and juice filter aids was 100% pure perlite. In the case of toothpaste, there are other components in the formulation which are company trade secrets.  In other studies where Imergard (perlite) was used to control arthropods, 100% pure perlite was used, the same concentration used in our studies.

  1. I suggest adding as an objective the investigation of the action mechanism of minerals on ticks. (Lines 81-82)

Authors’ response: Change made as suggested by adding “and mode of action” to this sentence to add this as an objective of the study.

  1. How the concentration per nymph could be calculated? It would be interesting to have a concentration per nymph, how much Celite each nymph received/was treated.

Was the concentration used similar/higher/lower than that used on other arthropods? (Lines 101-102)

Authors’ response: While calculating the concentration per nymph is something we were interested in pursuing, it is difficult as discussed earlier in our response to this reviewer. The amount of mineral on each tick is too small to be weighed with conventional equipment. Other alternative methods to calculating this is a project in itself. For other published studies using our minerals on arthropods, the concentration remained the same (100%) but the application methods were different and as the reviewer suggested, efficacy cannot be compared. But this is not just about the amount on each insect but where are the minerals on the insect.  We currently do not know the cite of action on the insect surface. So this is a complicated problem to solve which cannot be conducted in one publication and that could change from insect to insect, insect to tick and method of application to method of application.  

  1. The word "applied" may not be the best to describe that the Petri dish was "closed". (Lines 104-105)

Authors’ response: Change made as suggested. That part of the sentence now reads: “the top of the plate was used to cover the bottom plate and the two plate halves were sealed together with Parafilm”

  1. Were the trials repeated three times with ticks of different "infestations/collections/days/populations" (ie 3 replicas) or were there 40 replicas? (Lines 115-120)

Authors’ response: We have tried to make this clearer in the text by adding the number of ticks in each replicate and defining a replicate. Most replicates consisted of 3 treatment and 3 control Petri dishes but sometimes due to a lack of ticks we used 2 treatment and control Petri dishes or out of an abundance of ticks we used 4 Petri dishes.

  1. Please spell it out explicitly, as this is the first reference to the term. (Line 132)

Authors’ response: Change made as suggested.

  1. (CI) (Line 132)

Authors’ response: Change made as suggested.

  1. NCSU- Please spell it out explicitly, as this is the first reference to the term. (Line 139)

Authors’ response: Change made as suggested.

  1. I did not understand what the authors want to show with figure 1, it was not explained either in the text or in the figure caption. Does it show Celite's physical structure alone, without being on the tick? I don't understand what this adds to the work, or what relevant information it brings. I suggest that the authors explain better what the figure represents and what relevant information it brings to the work, or that it be removed. (Line 147)

Authors’ response: The purpose was to show the composition of Celite itself separate from how Celite appears in what is a typical position on the cuticle proper. Seems reasonable to try to show what we are applying, and the picture shows the actual structures produced by the diatoms. 

  1. This is already in the methodology, I think it is repetitive. (Lines 147-152)

Authors’ response: This has now been deleted to avoid repetition.

  1. Abbreviation for minutes is min, without the s. Remove the word "Percent" from the y-axis, leaving only the symbol in parentheses (%). (Lines 156-157)

Authors’ response: Change made as suggested.

  1. Remove response (Line 158)

Authors’ response: Change made as suggested.

  1. Italic (Line 158)

Authors’ response: Change made as suggested.

  1. Add the acronym in the statistic where it appears for the first time (163)

Authors’ response: Change made as suggested.

  1. Confusing, please improve. (Lines 164-165)

Authors’ response: Changed to (95% CI 77.44 to 85.86 min) to match the other examples.

  1.  

Authors’ response: Changed to match the others as suggested.

  1. This sentence does not read well, it is confusing, repetitive: "increased... increased". In addition, the increase in lethal time was not slightly, as there was a statistical difference. (Lines 165-166)

Authors’ response: The sentence has been changed to: “Higher age increased the time to death at the LT50 and LT90 by 1.2 and 1.1-fold, respectively.”

  1. Strikethrough text Line 168

Authors’ response: Change made as suggested.

  1. Please rephrase this sentence to improve the readability. (Line 170)

Authors’ response: Rephrased to: “To confirm mortality, all ticks were found to be dead at 24 h as well.”

  1. Indicate the units: days. (Line 178)

Authors’ response: Change made as suggested.

  1. interval (CI) (Line181)

Authors’ response: Change made as suggested.

  1. I think it is interesting to add results related to the observation or not of damage to the cuticle, since in the introduction it is said that this would be the mechanism of action of the minerals. (Line 183)

Authors’ response: This has now been added to the results.

  1. Figure 3A shows a dorsal view of the tick, right? So, I think that what is being visualized in the gnatosome is the chelicera and palps, not the hypostme. (Lines 186-187)

Authors’ response: You are correct. We have changed “hypostome” to “capitulum”.

  1. I believe that the authors are talking about figure 3C, since figure 3B shows only the idiosome of the tick, where it is not possible to visualize the hypostome. Even so, figure 3C seems to be from a dorsal view, so what I see are the chelicera and palps, not the hypostome. (Lines 187-188)

Authors’ response: This has been corrected to say Figure 3B and “hypostome” has been changed to “capitulum”.

  1. Wouldn't it be spiracles? (Line 190)

Authors’ response: Changed to “spiracular plate”

  1. Gnatosome, or palps and chelicerae (Line 195)

Authors’ response: Changed to “capitulum.”

  1. I think that this information should not be in the figure caption, but in the text, in the description of the results. (Lines 196-197)

Authors’ response: Change made as suggested.

  1. I suggest that figures 3 and 4 are merged, making a single board with the images. (Line 199)

Authors’ response: We separated the figures because Figure 4 focuses on the respiratory system only where images of the respiratory system of Ixodes scapularis nymphs have not been published before. Figure 4 results are the first evidence that our minerals might be affecting respiration, and we decided a separate figure was needed to emphasize this discovery. However, if the editor feels strongly that we need to combine the figures, then we will happily do so.  

  1. Strikethrough “obscured”, change to “obstructed” (Line 200)

Authors’ response: Change made as suggested.

  1. I think it would be better to present all the results of the Celite and Imegard bioassay/virulence together and then present the microscopy result, as it is in the methodology. (Line 202)

Authors’ response: The results have been reorganized as suggested.

  1. Strikethrough text-“unfed” (Line 203)

Authors’ response: Change made as suggested.

  1. Strikethrough text-“,” (Line 203)

Authors’ response: Change made as suggested.

  1. Add “)” (Line 205)

Authors’ response: Change made as suggested.

  1. Add “on unfed, 50–70 d old nymphs” (line 207)

Authors’ response: Change made as suggested.

  1. Same observations in relation to figure 2: abbreviation of minutes - min - and legend of the y axis - Mortality (%) (Line 208)

Authors’ response: Change made as suggested.

  1. Strikethrough text “response” (Line 209)

Authors’ response: Change made as suggested.

  1. Wouldn't "the difference in time of death was small"? Again, the numerical difference does not matter, what matters is the statistical significance. (Lines 215-216)

Authors’ response: The sentence was rephrased and now reads: “The age difference in time to death might reflect a lower water permeability across the cuticle and/or desiccation resistance of the tick in general in the older nymphs”

  1. I think this topic can be better discussed and explored, relating the maturity of the cuticle, since older nymphs should have a more mature, more developed cuticle, providing better protection against desiccation. (Line 217)

Authors’ response: This was added to the sentence to explore this point further “as older nymphs may have a more developed cuticle that provides better protection against desiccation.”

  1. It is important to add this information to the results. (Lines 243-244)

Authors’ response: This has now been mentioned in the results.

  1. Under the same conditions/concentrations? Strange, since normally insects are more susceptible to insecticides/acaricides than ticks. (Line 245)

Authors’ response: We have now deleted a part of the sentence as the treatment conditions in experiments with mosquitoes and filth flies were conducted differently and this made this an invalid statement. The sentence now reads “Furthermore, the ticks were not physically active after treatment suggesting abrasion was not a factor in cuticle damage.

  1. I did not understand this relationship of ticks not being physically active with the absence of damage to the cuticle. They may have remained immobile in order to reduce desiccation after damage to the cuticle. In fact, this behavior was mentioned by the authors in the above paragraph. (Lines 245-246)

Authors’ response:  The consensus view in the literature is that when insects are coated with diatomaceous earth, afterwards their movement and interaction with objects in their environment some of which could also be coated with DE, scratches the cuticle, damaging the cement coat of the cuticle and the lipid layer below. This causes dehydration.  If the ticks don’t move, this mechanism does not happen. 

  1. The statistical analysis showed that Imergard is more effective against I. scapularis nymphs than Celite. I don't think they are almost equally efficacious. (Lines 260-262)

Authors’ response: The word “slightly” has been deleted.

  1. Add “unfed nymphs of” (Line 282)

Authors’ response: Change made as suggested.

  1. Add “in laboratory conditions” (Line 282)

Authors’ response: Change made as suggested.

  1. I find it inappropriate to say this, as it is clear from their results that the mechanism of action has not yet been elucidated. I also suggest including in the conclusion your suggestions/observations related to the mechanism of action (Line 288)

Authors’ response: We have added the word “putative” before mechanical mode of action. This sentence has been added to the conclusion to include our observations on the mode of action, “The mode of action of these industrial minerals is not clear. No damage to the cuticle was observed.  The minerals appeared to obstruct openings to the respiratory system.”

Reviewer 3 Report

The authors give interesting information on the use of mechanical insecticides against the tick Ixodes scapularis. This information is not available, and it adds great value to the scientific community.
The experimental design is in accordance with the answers that the authors presented.
 The article is well done and well written and the results are well discussed.

the only remark is on the list of references:

many references need to be corrected, completed, and homogenized:

e.g., in reference 39: lack of the journal's name, volume, and the number of pages. In reference 40: the year of publication should be in bold and dot in the end instead of a semicolon...etc.

for this, I suggest that the author go through and revise the whole reference list,

Author Response

Reviewer 3:

  1. The authors give interesting information on the use of mechanical insecticides against the tick Ixodes scapularis. This information is not available, and it adds great value to the scientific community. The experimental design is in accordance with the answers that the authors presented. The article is well done and well written and the results are well discussed.

Authors’ response: Thank you for your review and comment! We appreciate it and are excited to have our research published in The Insects.

  1. The only remark is on the list of references: many references need to be corrected, completed, and homogenized: e.g., in reference 39: lack of the journal's name, volume, and the number of pages. In reference 40: the year of publication should be in bold and dot in the end instead of a semicolon...etc. for this, I suggest that the author go through and revise the whole reference list

Authors’ response: We have now gone through and corrected mistakes in the reference list. In regard to reference 39, there are no pages reported for this online publication, but I have added the DOI. For reference 40, the semicolon has been changed to a period. However, the instructions on The Insects website says not to bold the year of books, so I have left that the same. 

Round 2

Reviewer 2 Report

I would like to congratulate the authors for the study, highlighting the importance of the topic addressed and the need for alternative forms to chemical acaricides.

I think that the present study was important to answer basic and initial questions related to the effectiveness or not of Celite and Imegard against I. scapularis, under laboratory conditions. It would be interesting and extremely important to address in future studies tests related to concentration/dose of application, since this is fundamental for the feasibility of use, financial viability,... given that these are limiting and challenging topics for the development of acaricides, since ticks are generally more resistant than insects and need higher doses for their control. That is, not every product applied to insect control is viable for tick control.

One last suggestion for the paper:

Remove everything highlighted in yellow from the conclusion. The study did not assess anything that is highlighted, so I don't think it's prudent to have this in the conclusion.

In summary, the mineral-based products, Celite 610 and Imergard WP were found to be efficacious against unfed nymphs of I. scapularis in laboratory conditions, making them promising potential acaricides. This discovery could aid in reducing the number of cases of Lyme disease as well as other tick-borne diseases in North America. Celite and Imergard are naturally occurring, considered safe, have an almost unlimited shelf life, can be applied as a wettable powder with conventional spray equipment used in vector control, and are found in abundance in nature. The mode of action of these industrial minerals is not clear. No damage to the cuticle was observed. However, the minerals obstructed the openings to the respiratory system. It appears these industrial minerals could be suitable alternatives to the currently used synthetic acaricides and because of their putative mechanical mode of action used as an additional tool for vector control and to manage tick pesticide resistance.

Author Response

Second reviewer, second review comment:

I would like to congratulate the authors for the study, highlighting the importance of the topic addressed and the need for alternative forms to chemical acaricides.

I think that the present study was important to answer basic and initial questions related to the effectiveness or not of Celite and Imegard against I. scapularis, under laboratory conditions. It would be interesting and extremely important to address in future studies tests related to concentration/dose of application, since this is fundamental for the feasibility of use, financial viability,... given that these are limiting and challenging topics for the development of acaricides, since ticks are generally more resistant than insects and need higher doses for their control. That is, not every product applied to insect control is viable for tick control.

One last suggestion for the paper:

Remove everything highlighted in yellow (corresponding author marked section for removal in this response using "><") from the conclusion. The study did not assess anything that is highlighted, so I don't think it's prudent to have this in the conclusion.

In summary, the mineral-based products, Celite 610 and Imergard WP were found to be efficacious against unfed nymphs of I. scapularis in laboratory conditions, making them promising potential acaricides. >This discovery could aid in reducing the number of cases of Lyme disease as well as other tick-borne diseases in North America. Celite and Imergard are naturally occurring, considered safe, have an almost unlimited shelf life, can be applied as a wettable powder with conventional spray equipment used in vector control, and are found in abundance in nature.< The mode of action of these industrial minerals is not clear. No damage to the cuticle was observed. However, the minerals obstructed the openings to the respiratory system. It appears these industrial minerals could be suitable alternatives to the currently used synthetic acaricides and because of their putative mechanical mode of action used as an additional tool for vector control and to manage tick pesticide resistance.

Authors' response:  the information was removed exactly as suggested.